# On the Performance of Energy Criterion Method in Wi-Fi Transient Signal Detection

Ismail Mohamed [1], Yaser Dalveren [2], Ferhat Ozgur Catak [3,*] and Ali Kara [4]

1 Communication Engineering Department, College of Electronic Technology, Bani Walid, Libya; ismael.salem@yahoo.com
2 Department of Avionics, Atilim University, Kizilcasar Mahallesi, Incek Golbasi, Ankara 06830, Turkey; yaser.dalveren@atilim.edu.tr
3 Electrical Engineering and Computer Science, University of Stavanger, 4021 Stavanger, Norway
4 Department of Electrical and Electronics Engineering, Gazi University, Eti Mahallesi, Yukselis Sokak, Maltepe, Ankara 06570, Turkey; akara@gazi.edu.tr
* Correspondence: f.ozgur.catak@uis.no

**Abstract:** In the development of radiofrequency fingerprinting (RFF), one of the major challenges is to extract subtle and robust features from transmitted signals of wireless devices to be used in accurate identification of possible threats to the wireless network. To overcome this challenge, the use of the transient region of the transmitted signals could be one of the best options. For an efficient transient-based RFF, it is also necessary to accurately and precisely estimate the transient region of the signal. Here, the most important difficulty can be attributed to the detection of the transient starting point. Thus, several methods have been developed to detect transient start in the literature. Among them, the energy criterion method based on the instantaneous amplitude characteristics (EC-*a*) was shown to be superior in a recent study. The study reported the performance of the EC-*a* method for a set of Wi-Fi signals captured from a particular Wi-Fi device brand. However, since the transient pattern varies according to the type of wireless device, the device diversity needs to be increased to achieve more reliable results. Therefore, this study is aimed at assessing the efficiency of the EC-*a* method across a large set of Wi-Fi signals captured from various Wi-Fi devices for the first time. To this end, Wi-Fi signals are first captured from smartphones of five brands, for a wide range of signal-to-noise ratio (SNR) values defined as low (−3 to 5 dB), medium (5 to 15 dB), and high (15 to 30 dB). Then, the performance of the EC-*a* method and well-known methods was comparatively assessed, and the efficiency of the EC-*a* method was verified in terms of detection accuracy.

**Keywords:** RF fingerprinting; transient detection; energy criterion; Wi-Fi





## 1. Introduction

Today, several efficient wireless technologies have emerged to provide reliable communication in wireless networks. One of the important concerns in such networks is to prevent users within the network from conducting malicious attacks. Traditionally, this concern can be addressed by using upper-layer security mechanisms in wireless networks. However, due to the limitations in their implementation, physical-layer security methods provide an alternative and efficient means of improving the security of the networks [1].

Radiofrequency fingerprinting (RFF) was recently proposed as a promising physical-layer security method used in wireless networks. Basically, it identifies wireless devices in the network through their unique or distinctive features (so-called RF fingerprints) extracted from their analog signal waveform. Here, the uniqueness of the feature is related to the imperfection in the analog components of a device [2]. Several methods in RFF have subsequently been developed in the literature, as comprehensively presented in [3].

Typically, a system based on RFF consists of three main stages, namely, signal capturing and preprocessing, feature extraction, and device classification. However, there are some

major challenges in the development of RFF that need to be addressed. The first is related to device diversity and the number of signals captured per device [4]. The second is related to the data acquisition system where high- or low-end receivers are used. These types of receivers are complex, expensive, and memory-intensive because of the higher sampling rates while recording signals [5]. The third major challenge is to extract the subtle and robust features of the signals transmitted from wireless devices, as this directly affects the performance of device classification. To address this challenge, researchers have mainly used two different regions of the transmitted signals over the course of many years: (a) the steady-state/preamble region of the signals emitted from 802.11a OFDM Ethernet devices [6,7], UMTS user equipment (UE) devices [8,9], USRP transmitters/devices [10–12], the source network interface card (NIC) [13], Cisco devices [14,15], RFID transponders [16], and MIMO radios [17]; (b) the transient region of the signals emitted from radio transmitters [18–23], Bluetooth transceivers [24], wireless sensor nodes [25,26], and smartphones [27–29].

In practice, the steady-state region of a transmitted signal depends on the transmitter type. However, the instability in the steady-state region presents a significant drawback for extracting robust features. On the other hand, the transient region commonly appears in a wireless transmission of all transmitter types. When compared to the steady-state-based RFF, the transient-based RFF offers higher performance only if the transient region (signal) is precisely extracted. However, there are serious difficulties in extracting the transient signal from the transmission because of the channel noise and short duration of the transient signal. Consequently, the most important challenge in transient-based RFF can be attributed to detection of the transient starting point. In the literature, several methods have been proposed in order to efficiently detect the transient starting point [30–36].

## 1.1. Related Works

In order to detect the transient starting point, one of the earliest methods, known as variance fractal dimension threshold detection (VFDTD), was proposed in [30]. In the study, the efficiency of the method was tested by radio signals collected from eight different radio transmitters. Mainly, the method is based on the idea that the transient signal can be detected by means of the fractal dimension. In the first stage of the method, the fractal dimension is calculated for each part of the signal segmented by a sliding window. In the second stage, the transient starting point is detected by defining a threshold experimentally. The threshold value here corresponds to the mean of the fractal dimension of channel noise. In general, although the VFDTD method offers a high detection rate, it is computationally complex, and it requires a thresholding mechanism.

Another transient detection method, known as the Bayesian step change detection (BSCD), detects the transients by using a posterior probability distribution function as presented in [31]. In the study, radio transmissions captured from 30 different transmitters were used in order to test the efficiency of the method. Basically, the approach proposed in the method is based on the use of Higuchi's method [37] for calculating the variance of fractal dimension for successive portions of the signal. Since the variance of fractal dimension between two consecutive sequences is correlated to the probability density function, the maximum value of the function is defined as the transient starting point. In general, the method has a poor detection rate, and it is computationally complex. On the other hand, as an advantage, it does not need a thresholding mechanism.

Another existing method is phase detection (PD) presented in [32], where the instantaneous phase characteristics of the signal are calculated to detect the start point of the transient. In the study, Bluetooth signals collected from three different Bluetooth transceivers were used to evaluate the detection performance of the PD method. In the first step of the PD method, the instantaneous phase of the signal is calculated. In the following step, the calculated signal is unwrapped to remove the discontinuities. Then, the absolute value of each element in the unwrapped vector is calculated, and, for each successive portion, the variance of the phase characteristics is calculated. In the last step, the fractal trajectory is created by obtaining the difference of the phase variance. In order

to detect the start point of the transient, the first five elements in the fractal trajectory are typically compared with a threshold value. The transient starting point is determined when the considered elements of the fractal trajectory approach the threshold value. From a general perspective, the method is simple, and it has a high detection rate. Yet, an accurate threshold value is required in its proper implementation.

The Bayesian ramp change detection (BRCD) method, which is an improved version of the BSCD method, also provides a means of transient starting point detection as presented in [33]. In the study, 802.11b Wi-Fi signals collected from nine Wi-Fi radios were used in order to verify the detection accuracy of the BRCD method. Fundamentally, the start point of the transient is detected by a Bayesian change detector that estimates the time instant when the power of the transmission signal is increased. Although it can be a better option for transient starting point detection when compared to BSCD method, it still has computational complexity, which leads to a significant drawback.

Moreover, the study presented in [34] proposed the mean change point detection (MCPD) method to detect the transient starting point. In the study, Wi-Fi signals were collected at different SNR levels (6 to 30 dB) from six different wireless network cards, and the detection accuracy of the method was examined. Basically, the MCPD method calculates the difference between the statistics of the signal samples. The maximum value of the calculated differences then yields the starting point of the transient. Overall, it is simple but accurate, and it does not need a thresholding mechanism. However, its computational speed is an important concern.

Furthermore, the study presented in [35] proposed the permutation entropy (PE) and generalized likelihood ratio test (GLRT) detector for detecting the start point of the transient. In the study, GSM signals were collected from a mobile phone at different SNRs (0 to 25 dB) to test the detection accuracy of the method. Essentially, in its initial stage, the difference between the complexity of noise and signals is measured by utilizing the PE algorithm. In order to obtain the PE trajectory of the timeseries, a sliding window is used. Hence, the change point in the PE trajectory gives the start point of the transient which is determined by means of a GLRT detector. From a general point of view, the method has a high detection rate, especially at low SNRs. However, as a drawback, it has computational complexity.

A novel method that utilizes the energy criterion (EC) technique was recently proposed in [36]. The idea of the method is to characterize the arrival of a signal by a variation of its energy content. In this context, two methods are offered, namely, the instantaneous amplitude characteristics-based energy criterion method (EC-$a$) and the instantaneous phase characteristics-based energy criterion method (EC-$\varnothing$). In order to test the detection accuracy of the methods, a dataset consisting of Wi-Fi signals collected from a smartphone at different SNR levels ($-3$ to 25 dB) was created. Although both methods are simple and provide a significant improvement in the detection accuracy and computational speed when compared to other well-known methods, the EC-$a$ method offers slightly better performance than the EC-$\varnothing$ method under different noise conditions. Nevertheless, its performance depends on the $\vartheta$ factor that enables reducing the delaying effect of the negative trend in separating the signal from the noise part. For this reason, it needs to be properly selected according to the SNR levels.

As a summary, the existing transient starting point detection methods are listed in Table 1, where signal types along with the SNR levels, number of transmitters (devices), and advantages and disadvantages of the methods are listed.

**Table 1.** A summary of existing transient detection methods.

| Method | Signal Type | Number of Devices | SNR | Advantages | Disadvantages |
|---|---|---|---|---|---|
| VFDTD [30] | Radio | 8 (Radio transmitter) | NA | ▪ Detection rate is high | ▪ Computationally complex ▪ Needs a thresholding mechanism |
| BSCD [31] | Radio | 30 (Radio transmitter) | NA | ▪ Does not need a threshold value | ▪ Detection rate is poor ▪ Computationally Complex |
| PD [32] | Bluetooth | 10 (Radio transmitter) | NA | ▪ Detection rate is high ▪ Simple | ▪ Needs a threshold value |
| BRCD [33] | 802.11b Wi-Fi | 9 (Wi-Fi radio) | NA | ▪ Does not need a threshold value | ▪ Computationally complex |
| MCPD [34] | Wi-Fi | 6 (WLAN card) | 6 to 30 dB | ▪ Detection rate is high ▪ Does not need a threshold value ▪ Simple | ▪ Computational time is moderate |
| PE & GLRT [35] | GSM | 1 (Smartphone) | 0 to 25 dB | ▪ Detection rate is high ▪ Does not need a threshold value | ▪ Computationally complex |
| EC-*a* [36] | Wi-Fi | 1 (Smartphone) | −3 to 25 dB | ▪ Detection rate is high ▪ Does not need a threshold value ▪ Simple | ▪ Sensitive to $\vartheta$ factor |

### 1.2. Aim of the Study and Contributions

Fundamentally, the transient pattern varies according to the type of wireless device. Therefore, in order to reliably analyze the detection accuracy of transient starting point, it is necessary to increase the device diversity. Otherwise, unreliable results are expected to be obtained because of insufficient size and the limited variety of the data. This, in fact, constitutes an important concern that needs to be addressed.

As can be deduced from the discussion of the previous subsection, the EC-*a* method has a significant advantage in terms of computational complexity and the detection performance over the well-known methods. However, in [36], its performance was examined for a set of Wi-Fi signals captured from a particular Wi-Fi device. Therefore, a serious concern can arise regarding its performance when the device diversity is increased. In this context, this study is devoted to resolving this concern by assessing the efficiency of the EC-*a* method across a large set of Wi-Fi signals captured from various Wi-Fi devices. To this end, the sets including Wi-Fi signals captured from the smartphones of five brands were created. SNR levels between −3 to 30 dB were then added to the Wi-Fi signals in each set to evaluate the effect of SNR on the transient start detection performance of the well-known methods [30–32,34] in comparison with the EC-*a* method. Next, the performance of the considered methods was comparatively assessed using the created datasets. According to the comparison results, the efficiency of the EC-*a* method was verified. The block diagram shown in Figure 1 depicts the overall process for the presented study.

Briefly, the contributions of this article are twofold:

(a) This is the first report that studies the validity of the EC-*a* method using large sets of Wi-Fi signals captured from various Wi-Fi devices;

(b) By utilizing the large sets of Wi-Fi signals under different SNR levels, the transient start detection performance of the well-known methods is comparatively assessed for the first time in the literature.

The article is organized as follows: in Section 2, the data acquisition system used in this study is introduced, the system setup and signal capturing are presented, and then the signal preprocessing which is a crucial stage before the experiments is described. In

Section 3, the EC-*a* method is explained briefly. The experiments conducted to assess the transient starting point detection performance of the EC-*a* method along with the well-known methods [30–32,34] under different SNR levels are presented in Section 4. Conclusions are provided in the last section.

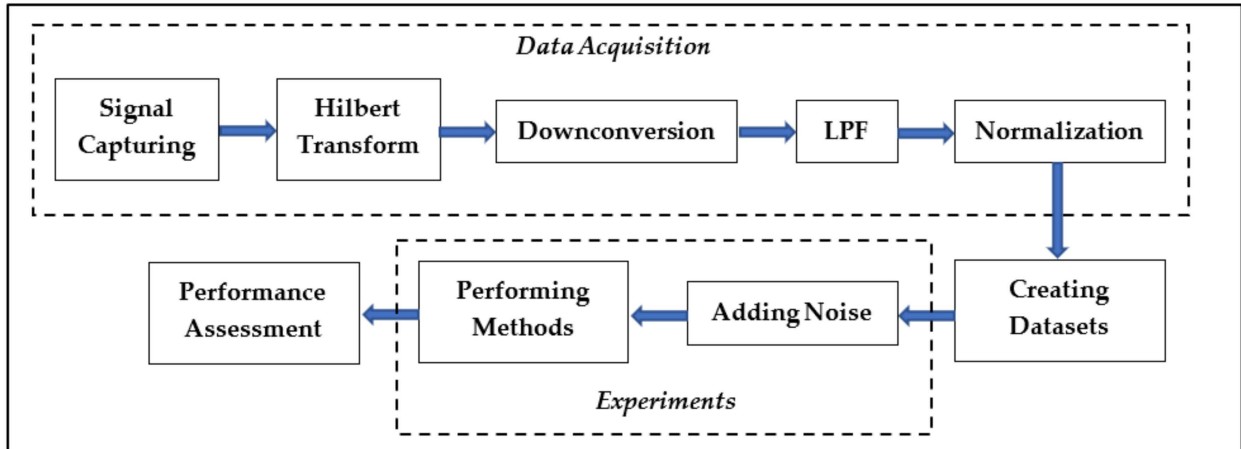

**Figure 1.** Overall process for the presented study.

## 2. Data Acquisition

### 2.1. System Setup and Wi-Fi Signal Capturing

Data acquisition was performed in an isolated laboratory environment located on the second underground floor of a nine-story building. Before the signal capturing process, the electronic devices around the acquisition system in the laboratory were switched off in order to avoid possible interferers.

The system setup used for data acquisition is shown in Figure 2. In order to capture Wi-Fi signals, a high-end receiver (Tektronix TDS7404 oscilloscope) was used in the system. A commercial Wi-Fi antenna connected to the oscilloscope was also used to collect the signals. The collected signals were then transferred to a computer for storage and further processing. The smartphone(s) used in the system was deployed 30 cm away from the antenna. During the process, the flight mode of the smartphone was activated to make sure that the undesired signals were not generated from the smartphone.

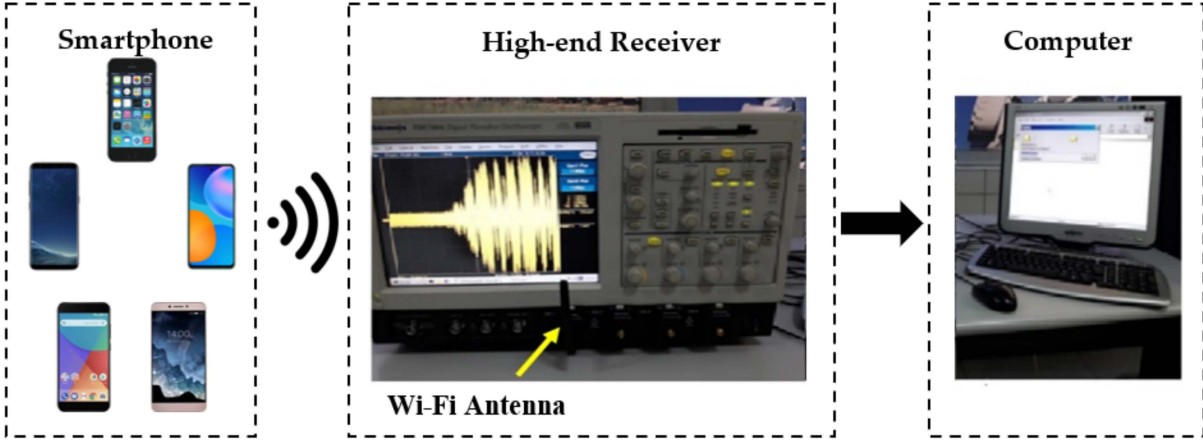

**Figure 2.** Data acquisition system.

The signals were captured at 20 Gsps sampling rate from five smartphones with different brands (Apple—iPhone 5, Samsung—Galaxy S8, Huawei—P Smart, Xiaomi—Mi A1, and LeEco—Le Max 2). In order to minimize the impairments on distinctive transient characteristics, down conversion was not used in the system. For each smartphone, sets of 100 Wi-Fi signals were recorded.

*2.2. Preprocessing*

After recording Wi-Fi signals, the undesired (spur) signals generated by the oscilloscope were removed. To do this, the captured intermediate frequency (IF) was first transformed into an analytical signal by using Hilbert transform (HT). Typically, for a real-valued discrete signal recorded in the time domain, $s_t(n)$, an analytical signal is written as

$$s(n) = s_t(n) + j\mathrm{H}\{s_t(n)\}, \tag{1}$$

where $\mathrm{H}\{s_t(n)\}$ denotes the Hilbert transform of the $s_t(n)$. The analytical signal can also be represented in terms of in-phase (*I*) and quadrature (*Q*) components as

$$s(n) = s_I(n) + js_Q(n). \tag{2}$$

The analytical IF signal was then down-converted to baseband using a complex exponential with $\omega_0 = 2.5$ GHz. Next, a low-pass filter (LPF) with 90 MHz cutoff frequency was applied in order to remove undesired frequency components. In the last step, the signal was normalized for scaling purposes.

As an example, Figure 3 shows the recordings of Wi-Fi signals obtained after the signal preprocessing stage. As can be deduced from the recordings, the smartphones had different signal characteristics in terms of two critical information such as "leading response" and "sharpness of transients". As shown in the figure, only the signals emitted from the smartphones such as Apple—iPhone 5 and Samsung—Galaxy S8 had a leading response. Leading response can be defined as a weak signal which is observed before the transient start [38]. This might cause a serious challenge while detecting the starting point of Wi-Fi transients. On the other hand, only the transients of the signals emitted from Xiaomi—Mi A1, Huawei—P Smart, and LeEco—Le Max 2 had sharp rising edges. As an advantage, this information is expected to increase the detectability of transient starting points. As an illustration, Figure 4 compares the recordings of Wi-Fi signals from Apple—iPhone 5 and Xiaomi—Mi A1 in order to clearly visualize the difference in the signal characteristics.

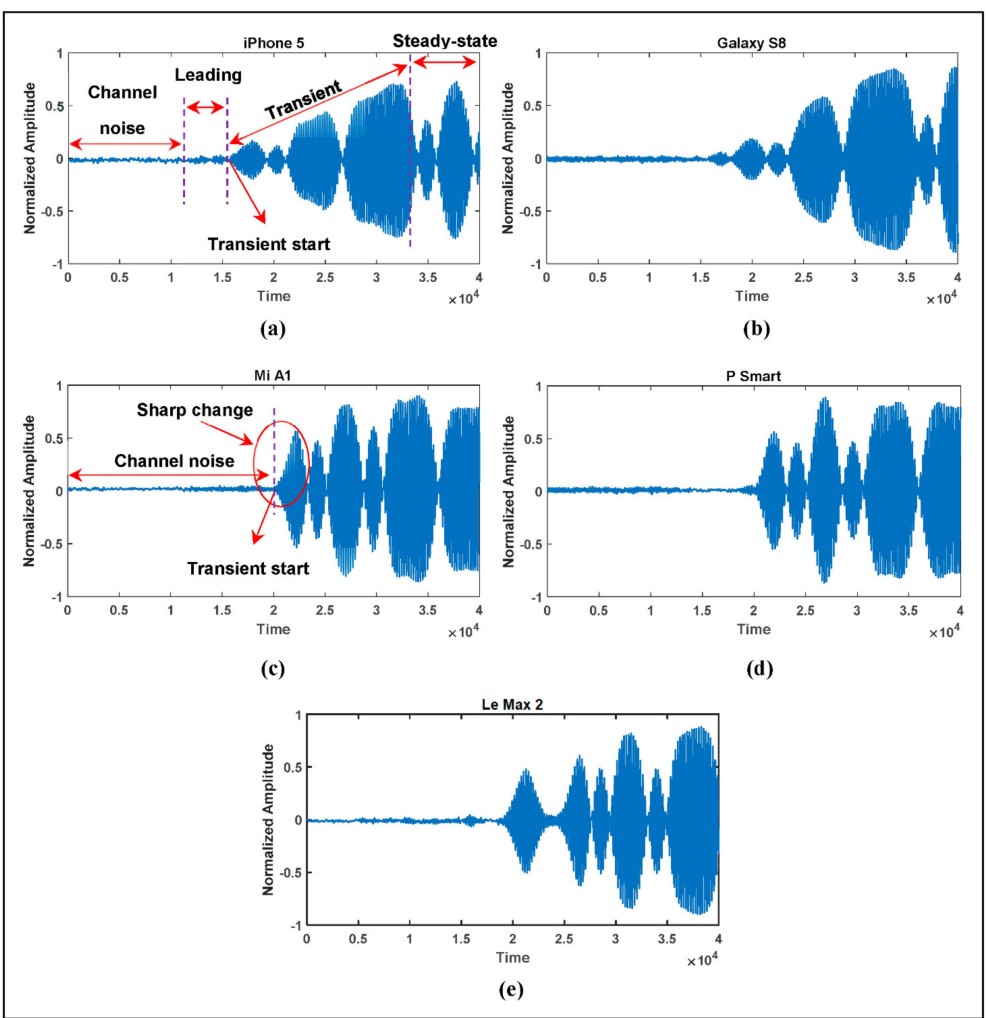

**Figure 3.** The recordings of Wi-Fi signals: (**a**) Apple—iPhone 5, (**b**) Samsung—Galaxy S8, (**c**) Xiaomi—Mi A1, (**d**) Huawei—P Smart, and (**e**) LeEco—Le Max 2.

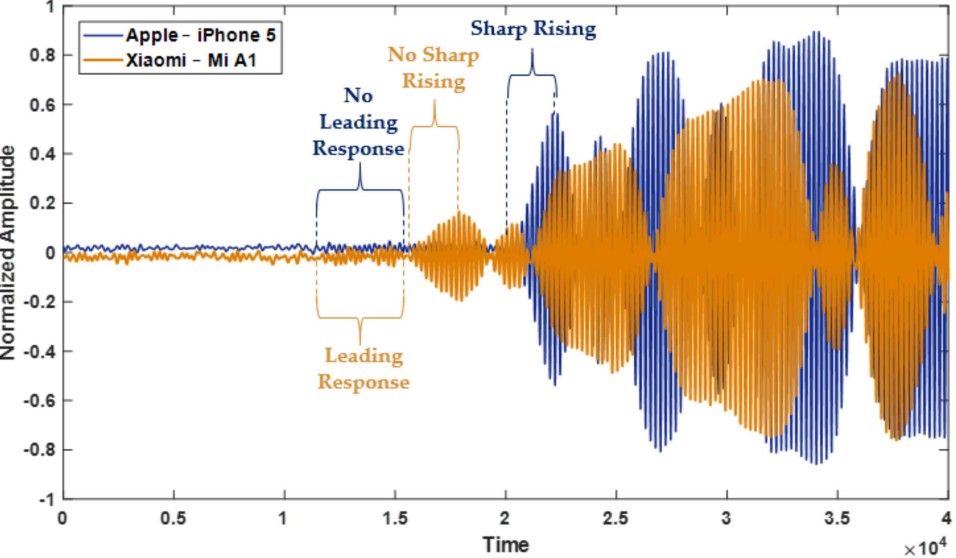

**Figure 4.** Comparison of Wi-Fi signals in terms of leading response and sharpness of transients.

### 3. Energy Criterion Method Based on the Instantaneous Amplitude Characteristics

The energy criterion (EC) is a basic method that is commonly used to detect the arrival times of the electrical signals. In this method, the arrival of an electrical signal is assumed to be characterized by a variation of its energy [39,40]. Simply, in order to implement the EC method, the energy of a sampled signal ($x$) needs to be defined as a cumulative sum of amplitude values.

$$E_i = \sum_{m=0}^{i} x_m^2, \ i = 1, \ldots, M, \tag{3}$$

where $M$ denotes the length of the signal. Here, the signal is then separated from its noise part by

$$E_i' = E_i - i\delta = \sum_{m=0}^{i} \left( x_m^2 - i\delta \right), \tag{4}$$

where $\delta$ denotes a negative trend, which is given as

$$\delta = \frac{E_M}{\vartheta \cdot M}, \tag{5}$$

where $\vartheta$ is a factor that reduces the delaying effect of $\delta$, and the parameter $\delta$ varies according to the total energy of the signal ($E_M$). Hence, the arrival time of the signals is considered as a global minimum of the calculated energy curve ($E_i'$).

Although the EC method has been applied for various applications so far, it was recently proven that it could also be an efficient means of transient starting point detection [36]. The idea is based on the use of instantaneous amplitude characteristics of the analytical signal to apply the EC method. In this context, a new method so called the EC-*a* method is proposed. The implementation of the EC-*a* method is simple and straightforward. In the first step, by referencing Equation (2), instantaneous amplitude characteristics of the analytical signal ($a(n)$) are calculated [14].

$$a(n) = \sqrt{s_I(n)^2 + s_Q(n)^2}. \tag{6}$$

In the second step, the calculated $a(n)$ characteristics of the analytical signal are utilized to determine the $E_i'$ curve using Equations (3) and (4). In the curve, the global minimum corresponds to the transient starting point. Nevertheless, the detection accuracy is strictly dependent on the value of the $\vartheta$ factor. Specifically, when denoised data are used, the global minimum value approaches the transient start if the $\vartheta$ factor is increased, such as $\vartheta \epsilon \mathbf{A}$, where $\mathbf{A} = [1, 2, 3, \ldots, 100]$. In other words, the value of the $\vartheta$ factor needs to be selected as high as possible for better detection accuracy. On the other hand, when noisy data are used, an optimum value of the $\vartheta$ factor needs to be defined by accounting for SNR levels in order to improve the detection accuracy as discussed in [36].

### 4. Experimental Results and Discussion

The main motivation to conduct the experiments was to assess the transient starting point detection performance of the EC-*a* method using sets of Wi-Fi signals captured from five Wi-Fi devices under realistic noise conditions. It was also intended to evaluate the effects of SNR on the transient detection performance of the EC-*a* method in comparison with well-known methods [30–32,34]. To achieve these goals, the channel noise captured in the data acquisition stage at different levels was randomly added into the Wi-Fi signals collected for each smartphone. Three datasets with different SNR levels were then created by varying SNR levels, which were defined as (a) low SNR ($-3$ to 5 dB), (b) medium SNR (5 to 15 dB), and (c) high SNR (15 to 30 dB). Next, the performance of the EC-*a* method and well-known methods was comparatively assessed in terms of detection accuracy. In the assessment, the absolute error metric was used as follows [36]:

$$\Delta p = |p_0 - p| / f_{s'} \tag{7}$$

where $f_s$ is the sampling frequency, $p$ is the estimated start of the transient, and $p_0$ is the actual start of the transient. The calculated transient starting point detection rates of the methods for the datasets under low SNR, medium SNR, and high SNR are listed in Tables 2–4, respectively.

**Table 2.** Transient starting point detection rates (%) at low SNR.

| Method | iPhone 5 | Galaxy S8 | P Smart | Le Max 2 | Mi A1 |
|---|---|---|---|---|---|
| VFDTD [30] | 93.1 | 95.3 | 98.6 | 97.6 | 99.4 |
| BSCD [31] | 75.7 | 72.5 | 98.3 | 91.3 | 97.0 |
| PD [32] | 89.8 | 93.5 | 95.6 | 95.4 | 95.6 |
| MCPD [34] | 97.1 | 97.7 | 99.1 | 99.1 | 99.5 |
| EC-*a* [36] | 99.5 | 98.6 | 98.7 | 99.4 | 99.6 |

**Table 3.** Transient starting point detection rates (%) at medium SNR.

| Method | iPhone 5 | Galaxy S8 | P Smart | Le Max 2 | Mi A1 |
|---|---|---|---|---|---|
| VFDTD [30] | 97.0 | 97.6 | 99.6 | 99.1 | 98.9 |
| BSCD [31] | 85.8 | 77.4 | 98.2 | 97.0 | 97.3 |
| PD [32] | 94.6 | 96.4 | 97.8 | 98.2 | 98.7 |
| MCPD [34] | 95.8 | 93.2 | 98.5 | 98.9 | 98.6 |
| EC-*a* [36] | 99.5 | 99.4 | 98.6 | 99.4 | 99.5 |

**Table 4.** Transient starting point detection rates (%) at high SNR.

| Method | iPhone 5 | Galaxy S8 | P Smart | Le Max 2 | Mi A1 |
|---|---|---|---|---|---|
| VFDTD [30] | 98.0 | 98.4 | 99.7 | 99.1 | 98.7 |
| BSCD [31] | 87.8 | 73.5 | 98.3 | 97.6 | 97.4 |
| PD [32] | 96.6 | 97.0 | 98.0 | 99.0 | 99.1 |
| MCPD [34] | 94.8 | 88.9 | 89.0 | 89.0 | 96.4 |
| EC-*a* [36] | 99.5 | 99.4 | 98.6 | 99.3 | 99.5 |

The detection accuracy of the methods can be analyzed in two aspects of Wi-Fi signals: the presence of leading response, and the sharpness of rising edge. For the signals emitted from Apple—iPhone 5 and Samsung—Galaxy S8 which have a leading response, it can be clearly observed from the results that the EC-*a* method had the highest detection accuracy at all SNR levels.

When the rising edge sharpness of the signals was considered, the EC-*a* method had the highest detection accuracy for the signals emitted from Xiaomi—Mi A1 and LeEco—Le Max 2 at all SNR levels. However, for the signals emitted from Huawei—P Smart, the VFDTD method seemed to have better detection accuracy at both medium and low SNR levels (99.6% and 99.7%, respectively), whereas the MCPD method had better detection accuracy at a low SNR level (99.1%), when compared to the detection accuracy of the EC-*a* method (98.7%). It is worth noting that the detection accuracy of the VFDTD method depends on a threshold value which should be properly determined for its implementation. Unless an optimum threshold value is properly determined, a significant reduction in the detection accuracy is expected. As for the MCPD method, which seems to have better detection accuracy at a low SNR level, its computational time could be a significant drawback for RFF as discussed in [36]. Evidently, due to the mentioned drawbacks of the VFDTD and MCPD methods, the EC-*a* method is still a strong candidate to detect the transient starting point owing to its acceptable detection accuracy (98.6%).

Transient starting point detection accuracies of the methods for Wi-Fi signals with
leading response and the signals with sharp rising edge at low and medium SNR levels are
shown in Figures 5 and 6, respectively. When the results are compared, the improvement in
the detection accuracy of well-known methods can be clearly observed. This suggests that
the sharp rising edge characteristics affect the detection accuracy of well-known methods,
especially at low and medium SNR levels.

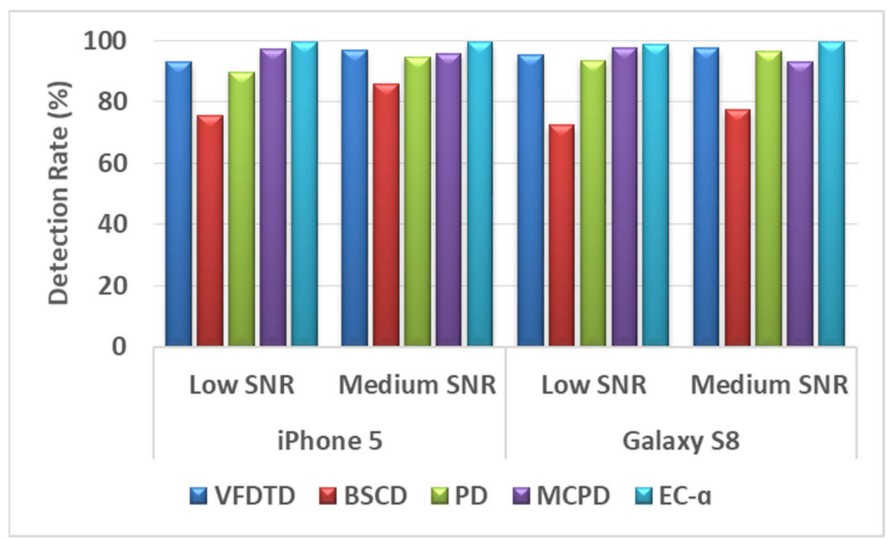

**Figure 5.** Transient starting point detection accuracy of the methods for Wi-Fi signals with leading
response at low and medium SNR.

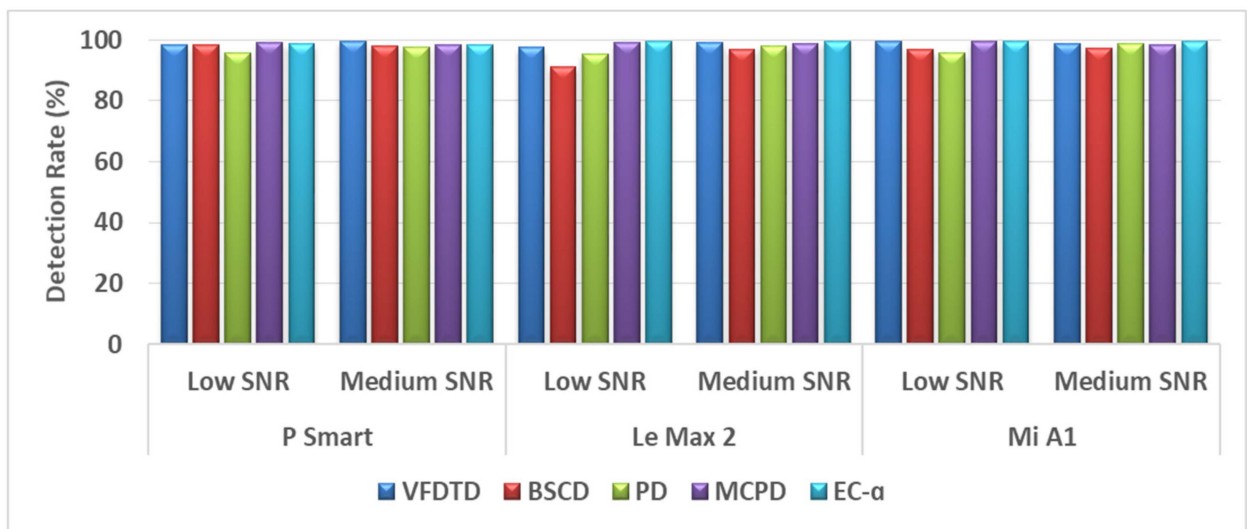

**Figure 6.** Transient starting point detection accuracy of the methods for Wi-Fi signals with sharp
rising edge at low and medium SNR.

Moreover, from a general perspective, the overall transient starting point detection
accuracies of the models at low, medium, and high SNR cases are also shown in Figures 7–9,
respectively. Using Tables 2–4, the mean value of transient starting point detection accuracy
for each method was calculated by

$$\overline{\Delta p} = \frac{\sum_{j=1}^{N} \Delta p(j)}{N}, \, j = 1, \ldots, N$$

where *N* denotes the number of devices used in the experiments.

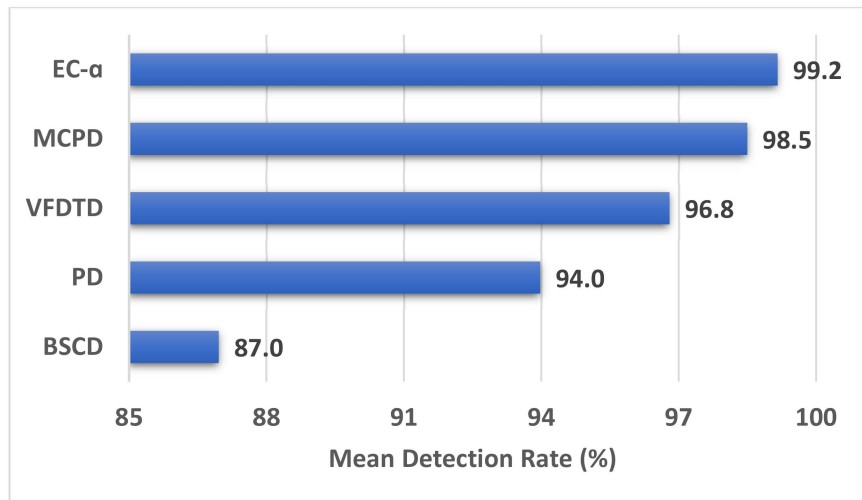

**Figure 7.** Overall transient starting point detection accuracy of the methods at low SNR.

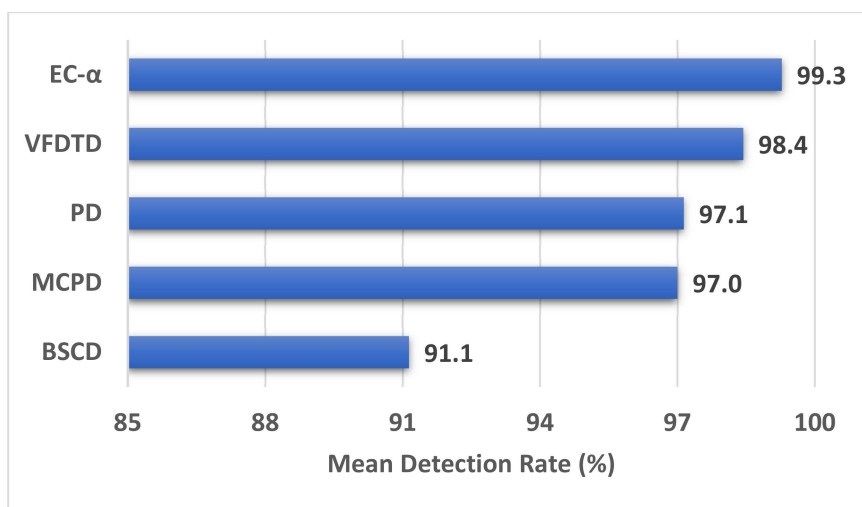

**Figure 8.** Overall transient starting point detection accuracy of the methods at medium SNR.

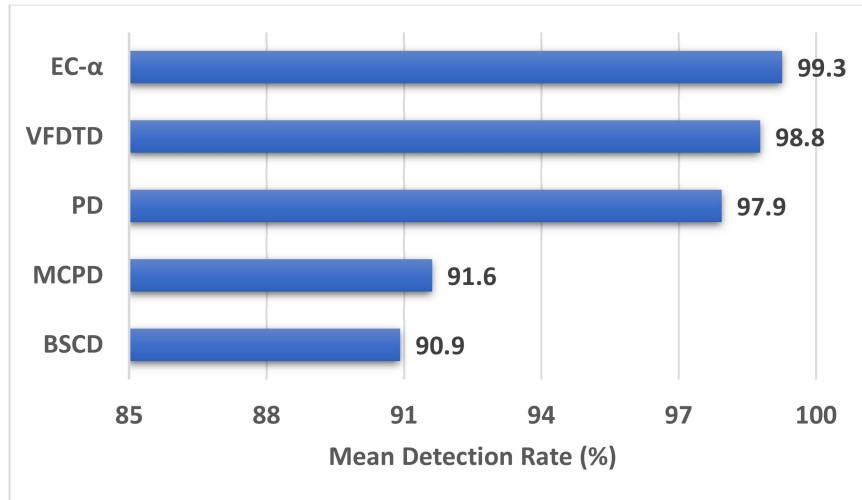

**Figure 9.** Overall transient starting point detection accuracy of the methods at high SNR.

As can be clearly seen from Figures 7–9, the EC-*a* method had the highest performance at all SNR levels. On the other hand, the performance of the VFDTD method seemed to

be consistent, as it provided similar mean detection rates at all SNR levels (about 98%). Another significant result is related to the performance of the MCPD method providing acceptable detection accuracy at low SNR (98.5%), while it had lower detection rates at other SNR levels (97% and 91.6% at medium and high SNR, respectively). At high SNR, it is worth noting that the PD method could be an option due to its acceptable detection accuracy (97.9%). However, at other SNR levels, the performance degradation can be explicitly observed (97.1% and 94% at medium and low SNR, respectively). From the results, it is also evident that the BSCD method had the lowest detection accuracy at all SNR levels, as expected.

Overall, the experimental results show the robustness of the EC-*a* method at all SNR levels. Obviously, this method provided almost similar detection accuracy at each SNR level. However, the transient detection performance of other methods was significantly affected by the variation of SNR levels. Therefore, it can be stated that the results achieved from this study are consistent with the results obtained in [36].

From the experimental results achieved in this study, Table 5 was created in order to compare the methods in terms of their detection accuracy at different SNR levels. Surely, the advantages and disadvantages of the methods that mentioned in Table 1 also need to be considered while comparing methods in terms of their applicability and usability in practice.

**Table 5.** Comparison of the transient start detection methods based on the experimental results.

| Method | Pros | | Cons | |
|--------|------|--|------|--|
| VFDTD | ☺ | Its performance is relatively stable when compared to other well-known methods. | ☹ | It has lower detection accuracy at low SNR when the signals having leading response characteristics are considered. |
| | ☺ | Could be a good candidate to the EC-*a* method in general. | | |
| MCPD | ☺ | Could be good candidate to the EC-*a* method at low SNR. | ☹ | Its performance tends to decrease when the SNR level is increased. |
| PD | ☺ | At high SNR, due to its acceptable performance, it can be an alternative to VFDTD method. | ☹ | Its performance tends to decrease when the SNR level is decreased. |
| BSCD | ☺ | Could be used for the signals with sharp rising edge characteristics. | ☹ | In general, it has poor detection accuracy. |
| EC-*a* | ☺ | Overall, it has the highest performance at all SNR levels. | ☹ | Slight degradation in its performance may be observed for the signals with sharp rising edge characteristics. |
| | ☺ | Its performance is robust to SNR levels. | | |

On the other hand, in this study, the measurements were only given for one type of device (smartphone) manufactured by five different brands, and the performance of the methods was comparatively assessed. It is believed that there are still opportunities to assess the transient detection performance of the EC-*a* method and the well-known methods so that a broader view of performance can be outlined. In this context, one of the possible options is to vary the smartphone models of each brand. However, the signals transmitted from different smartphone models of the same brand are expected to have similar characteristics. In this case, only slight and ignorable differences can be observed between the signal waveforms. Therefore, similar detection accuracies are

expected to be obtained. Yet, further studies need to be conducted in order to support this argument. Furthermore, another possible option is to extend the measurements using the signals transmitted from different types of devices such as radio transmitters [30,31], Wi-Fi radios [33], and WLAN cards [34]. In fact, all these research directions constitute future works that will be carried out by the authors in the near future.

## 5. Conclusions

The experimental study presented in this article comparatively assessed the transient starting point detection performance of the EC-*a* method and well-known methods by using large sets of Wi-Fi signals captured from five Wi-Fi devices under different SNR levels ($-3$ to 30 dB). The detection accuracy of the methods was analyzed in two aspects of Wi-Fi signals: (a) presence of leading response, and (b) sharpness of rising edge. One of the important findings is that the EC-*a* method had the highest detection accuracy for the signals having a leading response. Another finding is related to the rising edge sharpness of the signals in which the EC-*a* method still provided acceptable detection rates although the VFDTD method and the MCPD method could be alternatives to the EC-*a* method because of their accuracy, especially at low and medium SNR. However, for an efficient RFF, it is necessary to account for the drawbacks of the VFDTD and the MCPD methods such as the required threshold mechanism and higher computational time.

It can be concluded that the EC-*a* method is robust and provides accurate detection performance at all SNR levels. To the extent of our knowledge, the performance of the EC-*a* method was herein validated for the first time using large sets of Wi-Fi signals captured from various Wi-Fi devices. Furthermore, the assessment of transient start detection performance of the well-known methods with large sets of Wi-Fi signals under different SNR levels is another contribution of this study.

**Author Contributions:** Data curation and methodology, I.M.; investigation, I.M., Y.D. and F.O.C.; conceptualization and validation, Y.D. and A.K.; formal analysis and visualization Y.D. and F.O.C.; writing—original draft preparation, Y.D.; writing—review and editing, and supervision, A.K. All authors read and agreed to the published version of the manuscript.

**Funding:** This research received no external funding.

**Institutional Review Board Statement:** Not applicable.

**Informed Consent Statement:** Not applicable.

**Data Availability Statement:** The data presented in this study are available on request from the corresponding author.

**Conflicts of Interest:** The authors declare no conflict of interest.

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
