# Peer review of "On the Performance of Energy Criterion Method in Wi-Fi Transient Signal Detection"

_electronics, doi:10.3390/electronics11020269_

Round 1
Reviewer 1 Report
The authors evaluate the performance of the Criterion based on the instantaneous amplitude characteristics method for a set of Wi-Fi signals captured from a particular Wi-Fi device brand.
- Authors are suggested to add an introductory figure in section 1.
- After figure 2, a comparative graph would explain the things in a better way.
- Section 4 is week and requires more elaborative evaluation results in form of graphs.
- Simulation parameters must be presented and the logic for the selected parameters should be discussed.
Reviewer 2 Report
This paper is an experimental study of detection of transient epochs in RF signals for fingerprinting. The authors first review the proposed methods in the literature, then report on an experimental study and performance of EC-alpha, the detection of the signal start by measuring the energy of the received signal (with a threshold).
The results in the paper give evidence that EC-alpha has superior performance to other more complex methods. However, the measurements have only been given for one type of device and frequency (smartphones, 2.5 GHz). Clearly, it had been more interesting to extend the measurements to other devices in the same or in other frequency bands, so that a broader view of performance could be outlined. Overall, the paper is clearly organized and written, contains enough (but not many) experimental results, and is interesting for engineers in the field.
Reviewer 3 Report
In this manuscript, the authors can detail the following in the main text:
Abstract: In the development of Radio Frequency Fingerprinting (RFF), one of the major challenges 14 is to extract subtle and robust features from transmitted signals of wireless devices for an accurate 15 device classification. To overcome this challenge, the use of transient region of the transmitted sig-16 nals could be one of the best options. For an efficient transient-based RFF, it is also necessary to 17 estimate the transient region of the signal accurately, precisely. Here, the most important difficulty 18 can be attributed to the detection of transient starting point. Thus, several methods have been de-19 veloped to detect transient start in the literature. Among them, the Energy Criterion method based 20 on the instantaneous... in the first sentence of the abstract, can you mention what does accurate device classification mean,? just the investigation is done considering only one type of wifi device. The transient region may have different response for different companies of wifi devices, can you disucss more on this respect?. As you mention the detection of the transient starting point is a challenge, and you may devote one section in the main text just to describe this effect and the application of well- known methods, also, considering what type of wifi devices have been used. You propose emphasizing the use of an already published method, and you compare it with similar ones; Can you quantify the suitability of each method in a comparative table? and Can you discuss what other type of wifi devices can be used?
In the main ext: Can you detail the different values of SNR when you mention the following sentence?
Wi-Fi signals are captured from smartphones of five brands, for wide range of sig-27 nal-to-noise ratio (SNR) values (‒3 to 25 dB).
You are encouraged to detail the content or emphasize what you are trying to said when citing more than five references at the same time, as done in the following sentences; ... To ease this challenge, re-54 searchers have mainly used two different regions of the transmitted signals over the 55 course of many years: a) Steady-state/Preamble region [6–17], b) Transient region [2,18– 30]. In practice, the steady-state region of a transmitted signal depends on the transmitter 57 type. ... detailing the main keywoerd in each reference may help to understand the different issues on the transmitted signals and the corresponding regions.
Can this paper be recommended as a review? since you clearly mention in the following sentence that you are revisiting the application of one method:
- a) This is the first report that studies the validity of the EC-? method by using the 173 large sets of Wi-Fi signals captured from various Wi-Fi devices.
Before describing Section 2. Data Acquisition, can you summarize the contents of the manuscript?, basically emphasizing the main objective of the remaining sections of your paper.
Table 2. Transient starting point detection rates (%) at low SNR. ... As well as tables 3 and 4 detecting medium and high rates...Can you describe the usefulness of this classification? are there other ways to identify the estimation of the starting point and the transient effects in general?
In the last sentence of section 4 you duplicate achieved: Therefore, it can be stated that the achieved results achieved from 328 this study are consistent with the results obtained in [37].
Round 2
Reviewer 1 Report
Authors have addressed the reviewer comments sincerely.